# Micromachined mmWave 28.0/38.0 GHz MIMO antenna loaded with frequency selective surface for gain enhancement and SAR analysis for future wireless applications

**Manish Sharma[1], Bhaskara Rao Perli[2], Geetanjali Singla [3], Tathababu Addepalli[4], Sivasubramanyam Medasani[5], B. Satya Sridevi[6], Tanweer Ali [7]***

**1** Department of Electrical, Electronics and Communication Engineering, Galgotias University, Greater Noida, Uttar Pradesh, India, **2** Department of ECE, St. Ann's College of Engineering and Technology, Chirala, India, **3** Department of Electronics and Communication Engineering, Thapar Institute of Engineering and Technology, Patiala, Punjab, India, **4** Department of ECE, Aditya University, Surampalem, India, **5** Department of CSE, K. S. School of Engineering and Management, Bengaluru, India, **6** Department of ECE, Aditya University, Surampalem, India, **7** Manipal Institute of Technology, Manipal Academy of Higher Education, Manipal, India

* tanweer.ali@manipal.edu

## Abstract

The presented work describes the four-port multiple-input multiple-output (MIMO) antenna designed on Rogers substrate with a thickness of 0.787 mm. The core of the MIMO antenna includes a hexagonal-patch etched by a rectangular slot and an etched hexagonal-ring in full-ground, which are printed on opposite surfaces, generating measured millimeter-wave (mmWave) bandwidth of 26.45 GHz-29.27 GHz and 37.04 GHz-39.12 GHz. The dielectric material is micromachined, and the four-port radiating elements maintain isolation of more than 20.0 dB with an overall size of 17.0 mm × 22.0 mm × 0.787 mm. The novel 11 × 11 hexagonal-ring frequency-selective-surface (FSS) is placed below the MIMO antenna, enhancing the peak-gain by 5.16 dBi with a size of 42.50 mm × 42.50 mm printed on Rogers5880 0.787 mm thickness. The MIMO antenna also features good diversity-performance with $ECC_{mmWave-FSS} < 0.18$, $DG_{mmWave-FSS} > 9.995$ dB, $TARC_{mmWave-FSS} < -4.76$ dB, and $CCL_{mmWave-FSS} < 0.30$ b/s/Hz. The cumulative features with dual millimeter wave bands, enhanced peak-realized gain, suppressed back-lobe radiations, and good diversity performance make the proposed MIMO antenna loaded with FSS suitable for 5G, satellite-communication, IoT, and smart cities applications. The MIMO antenna loaded with FSS is also subjected for SAR analysis for input power of 50 mW and 500 mW.

## 1. Introduction

The fine spatial-resolution and short wavelength in the millimeter-wave range with features such as ultra-fast transfer of data with low latency has resulted in massive

**Data availability statement:** All relevant data are within the manuscript.

**Funding:** The author(s) received no specific funding for this work.

**Competing interests:** The authors have declared that no competing interests exist.

applications in 5G/6G wireless communication where they are used as high capacity back-haul links. The wireless communication at 5G/6G communication also demands multiple-port compact antenna which can be largely used for 28.0 GHz/38.0 GHz applications. The significance of the antenna design with encountering trade-offs is discussed.

A stepped-shape radiator placed orthogonally with four-port input and commonly-connected L-shaped ground uses RT-5880 dielectric with dimensions of $20.48 \times 20.48$ mm$^2$, generating 28.0 GHz operational-bandwidth of 25.21 GHz-32.34 GHz and records port-to-port isolation of more than 20.0 dB [1]. A two-port multiple-input-multiple-output (MIMO) antenna resonates at 27.80 GHz with a peak-gain of 5.42 dBi (size: $15 \times 30 \times 0.254$ mm$^3$), which includes a slotted rectangular-patch and full-ground with an etched rectangular-slot placed below the patch [2]. A multi-iterated four-port MIMO antenna placed orthogonally sharing a defected-ground-structure (DGS) generates resonance at 28.0 GHz, which is suitable for next-generation IoT networks [3]. A machine learning approach is reported in designing of 28.0 GHz MIMO antenna where the antenna gain is predicted by using ML modes and evaluating R-squared, Variance-score, mean-absolute-error, and root-mean-square (RMS) error [4,5,6]. A coaxial feed two-port MIMO antenna placed orthogonally achieves isolation by placing a parasitic-patch between the radiating-elements [7], a two-port MIMO antenna generating a bandwidth of 36.0 GHz-40.0 GHz, and additionally, the loading of meta-surface converts linear to circular-polarization [8,9]. A two-element array placed adjacent to each other is printed on Rogers-family with dimensions of $20.0$ mm $\times 20.0$ mm, generating a 28.0 GHz bandwidth of 25.20 GHz-29.40 GHz with a peak-gain record of 11.50 dBi [10]. A circular-slotted patch placed orthogonally with a corrugated connected-ground occupies a space of $30.0$ mm $\times 30.0$ mm and is designed for 28.0 GHz mmWave applications [11]. A four-port MIMO antenna with a unique arrangement is analyzed by studying an equivalent-circuit model, and also the specific absorption rate (SAR) is calculated with values less than 1.60 W/kg [12,13,14]. The references [1–5,7–13] discuss two-port/four-port MIMO antennas with single-band generation with resonance either centered at 28.0 GHz or 38.0 GHz. However, the dual-band (28/38 GHz) is also discussed [6]- [15], which uses a different technique and isolation method.

Also, a two-port MIMO antenna with dual-band of operation (28 GHz/38 GHz) utilizes dielectric resonators [16], and a $1 \times 4$ MIMO antenna resonating at 28.0 GHz and 38.0 GHz are placed adjacent to each other, achieving isolation of more than 24.0 dB [17]. This MIMO antenna calculates SAR for 1g/10g of the tissue model, where values are within the limits with input power of 25.0 mW [17]. Also, tri-band MIMO-antenna with centered resonance frequency of 28.0 GHz/35.0 GHz/38.0 GHz fabricated on Rogers substrate utilizes an isolated pair of metamaterial structure for matching of the impedance and also prevents side-reflections [18,19,20,15,21–25]. A dual-band two-port MIMO antenna [26] uses a parasitic-patch placed between the adjacent radiators to enhance the isolation, and a micromachined dielectric substrate with four-radiating patches with each array of two elements [27] generates 28.0 GHz/38.0 GHz narrow bands. The challenges, such as obstacles including rain,

path-of-travel, and advantages including higher-data-rate transmission, low latency features, are elaborated in [28] with millimeter waves finding their applications in satellite communication, peer-to-peer (P2P) communication, 5G stations, and mobile communication. The loading of frequency-selective-surface (FSS) [29–35]-[35] ensures the enhancement of the peak-gain by reflecting the back-lobes and thus achieving narrow-beams. A 3 × 3 matrix acts as a reflector utilizing FR4 substrate of thickness 1.60 mm to ensure the rise in peak-gain [29], and a double-sided FSS is designed in accordance with the resonance frequency of 28.0 GHz with absorption of around 90% [30]. Super-wideband FSS is reported, which is useful for bandwidth ranging between 3.0 GHz and 40.0 GHz [31–35], and is used with a super-wideband antenna, enhancing the gain. Also, millimeter-wave MIMO antenna [36–38], [39,40] using the reflector placed below antenna at λ/4 achieves constructive interference which enhances the overall peak-realized-gain of the antenna. The signal power characteristics is commonly used however, the fundamental limits of covert communications are discussed and the random-access problem in MIMO system with Rayleigh fading channels is also discussed in detail [41,42].

This manuscript presents a dual-band, high-gain, four-port mm-wave MIMO antenna designed for 28/38 GHz 5G applications. The detailed and comprehensive SAR analysis is also carried out to ensure safety and compliance. The proposed antenna demonstrates significant potential in meeting the multifaceted demands of next-generation wireless communication systems. The key contributions of the work are summarised as follows:

1. This MIMO antenna is designed with optimized compact dimensions (≤ $1\lambda_0$) and engineered DGS to ensure dual-band operation, high isolation, and low envelope-correlation-coefficient (ECC), suitable for 5G/mmWave applications. The geometry fosters current confinement and radiation diversity, addressing typical MIMO challenges like pattern correlation and mutual coupling.

2. A compact four-element multiple-input multiple-output (MIMO) antenna is proposed in this paper that addresses the low gain and bandwidth limitations of microstrip patch antennas. The proposed design has simple planar geometry, facilitates cost-effective manufacturing, and exhibits dual-band performance over 24 GHz/38 GHz bands, suitable for mm-wave 5G applications.

3. By strategically placing the slits in the hexagonal radiating elements and adjusting the feed of the designed antenna, dual-band performance is demonstrated, demonstrating substantial improvement in impedance bandwidth at each band.

4. To mitigate coupled currents in the MIMO configuration, adjacent antenna elements are oriented with 900 rotational symmetries supplemented by two additional optimized rectangular-shaped decoupled slots in the ground plane, yielding improved isolation up to 32 dB without requiring complex design geometry.

5. The antenna's performance is validated through experimental measurements of typical performance metrics of MIMO systems, which include S-parameters, diversity gain (DG), envelope correlation coefficient (ECC), and channel capacity loss (CCL). The results demonstrate strong agreement with simulation data, confirming the workability of the design.

The article is structured as follows: Section II represents the design evolution and the relevant parametric studies of the proposed single-element MIMO antenna design. Next, Section III expands the concept towards the design of a two-element and four-element MIMO arrangement, discussing its isolation mechanism and far-field performance. Section IV presents the fabricated prototype and compares the final measured results with the simulation results, followed by a conclusion and future scope of research in Section V.

## 2. The design methodology of a single-port dual-band mmWave antenna

The illustration of the millimeter-wave antenna resonating at 28.0 GHz and 38.0 GHz is shown in Fig 1. The isometric or the 3D-view is shown in Fig 1(a), where both planes of the substrate are printed with a novel radiating patch and defected

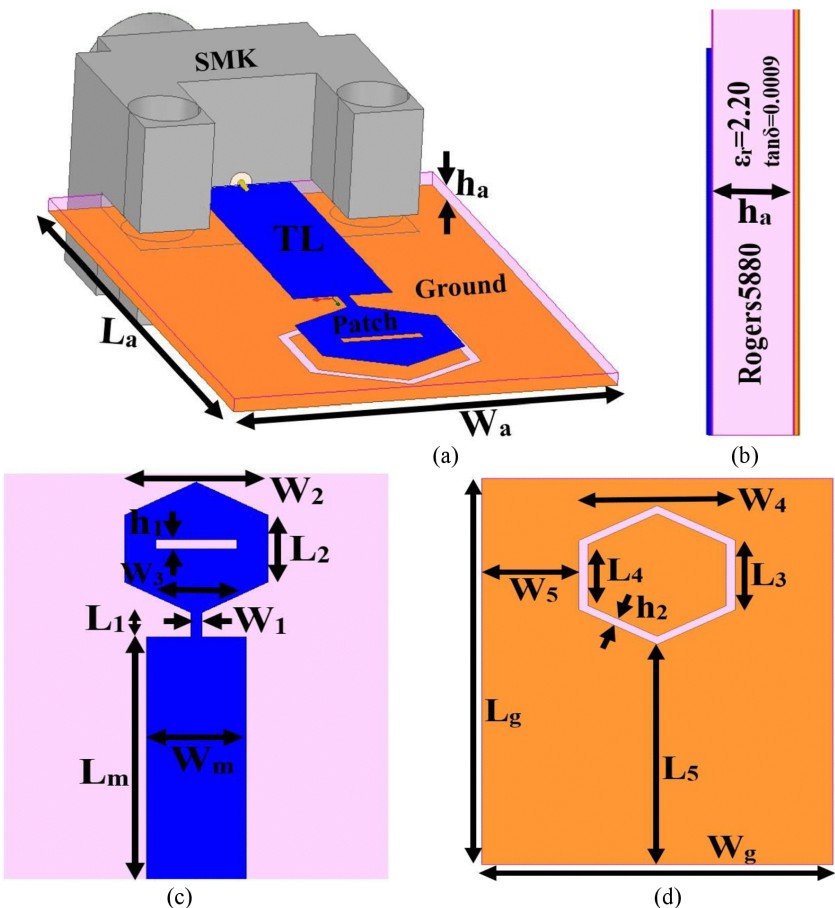

**Fig 1. Single mmWave antenna.** (a) The isometric-view (b) Side-view of dielectric details (c) Patch dimension details (d) Ground dimension details.

ground. The dimension of the antenna corresponds to $W_a \times L_a$ mm² with a radiating patch connected to the transmission-line (TL). The ground is printed on the opposite surface, which is defective as observed from Fig 1(a). The thickness of the substrate ha mm, and the antenna is fed by a simulation modeled 67.0 GHz SMK-type connector. Fig 1(a) shows the side-view of the antenna where Rogers5880 substrate with permittivity and loss-tangent corresponds to 2.20 and 0.0009. The metallic patch and ground are of a thickness of 35 μm. Fig 1(c) shows the radiating-patch view with marking of optimal dimensions. The transmission line (TL) of dimension $W_m \times L_m$ mm² is connected to a quarter-wave-transformer (QWT) of dimension $W_1 \times L_1$ mm². Also, the radiating patch, which is a hexagonal geometry, consists of a side length $L_2$ mm and a distance between the two opposite edges of $W_2$ mm. Moreover, the hexagonal geometry is etched by a rectangular slot of dimension $h_1 \times W_3$ mm².

The opposite surface of the substrate is printed with a full-ground plane of dimension $Wg \times Lg$ mm², as shown in Fig 1(d), and just behind the hexagonal-radiating patch, a hexagonal ring is etched in the ground with side-length $L_3$. The distance between the two opposite edges is $W_4$ mm. The inner-ring side-length corresponds to $L_4$ mm with a thickness of the ring $h_2$ mm. The hexagonal ring is placed at a distance of $W_5$ mm from the edge (length of the substrate) and by $L_5$ mm (width of the substrate). The dimensions marked in Fig 1 are optimized and are given in Table 1.

The dual-band with center-resonance frequency at 28.0 GHz and 38.0 GHz generated by a single-port antenna shown in Fig 1 is achieved by applying the stepwise transformation of the patch antenna with circular radiator and full-ground.

**Table 1. Optimal dimensions of 28.0/38.0 GHz single-port antenna.**

| Parameter | Dimension in mm | Parameter | Dimension in mm | Parameter | Dimension in mm |
|---|---|---|---|---|---|
| $W_a$ | 7.00 | $W_3$ | 1.50 | $L_3$ | 1.80 |
| $L_a$ | 10.0 | $W_4$ | 3.10 | $L_4$ | 1.60 |
| $h_a$ | 0.787 | $W_5$ | 1.95 | $L_5$ | 5.70 |
| $W_1$ | 0.20 | $L_1$ | 0.60 | $h_1 = h_2$ | 0.20 |
| $W_2$ | 2.60 | L2 | 1.50 | $W_m$ | 1.80 |
| $L_m$ | 5.50 | $W_g$ | 7.00 | $L_g$ | 10.0 |

The final version of the antenna shown in Fig 1 passes through four-step modifications, which are discussed in Fig 2. Step 1 consists of a circular-patch antenna and a full-ground plane shown in Fig 2(a) with very poor matching of impedance matching and hence, no radiations take place.

Hence, the need for modification arises in the form of Step 2 shown in Fig 2(a) with the addition of a quarter-wave transformer (QWT) between the transmission-line (TL) and the radiating-patch.

$$f_{LC} = \frac{c}{\lambda_{LC}} = \frac{7.20}{\left(W_2 + \frac{W_2}{2} + L_m + L_1\right)} \tag{1}$$

Where $W_2$ is the height of the hexagon, $W_2/2$ is the effective radius, $(L_m + L_1)$ is the length of the microstrip, and the lower cut-off frequency. The values of $a$ and $a_1$ are evaluated as

$$W_2 = \frac{6 \times a}{4\pi} = \sqrt{3} \times L_2 \tag{2}$$

This transformation shows the matching of the impedance around 37.0 GHz, generating a −10.0 dB bandwidth of 35.254 GHz-38.460 GHz ($f_r = 36.882$ GHz, S11 = −21.51 dB) noted from Fig 2(d). The Step3 iteration involves replacing the circular-patch with a hexagonal-geometrical patch, which shifts the resonance from 36.882 GHz to 37.90 GHz, closer to the 38.0 GHz n260 mmWave band. The hexagon-patch dimensions are calculated from Equations (1) and Equation (2). The final iteration, Step 4, shows the etching of a rectangular-slit on the hexagonal patch and cutting the hexagonal ring in the ground shown in Fig 2(a). This generates an additional 28.0 GHz band with improved matching of impedance at 38.0 GHz.

The length ($L_a$) and width ($W_a$) of the proposed antenna are evaluated from Equations (3) and Equation (4) given below.

$$L_a = \frac{C}{2 \times f_{LC}\sqrt{\frac{\varepsilon_r + 1}{2}}} \tag{3}$$

$$W_a = \sqrt{3} \times L_2 \tag{4}$$

The final version of the dual-band mmWave antenna is also realized by extracting the passive components' values (R, L, C) using Equation (5), Equation (6), and Equation (7).

$$f = \frac{1}{2\pi\sqrt{LC}} \tag{5}$$

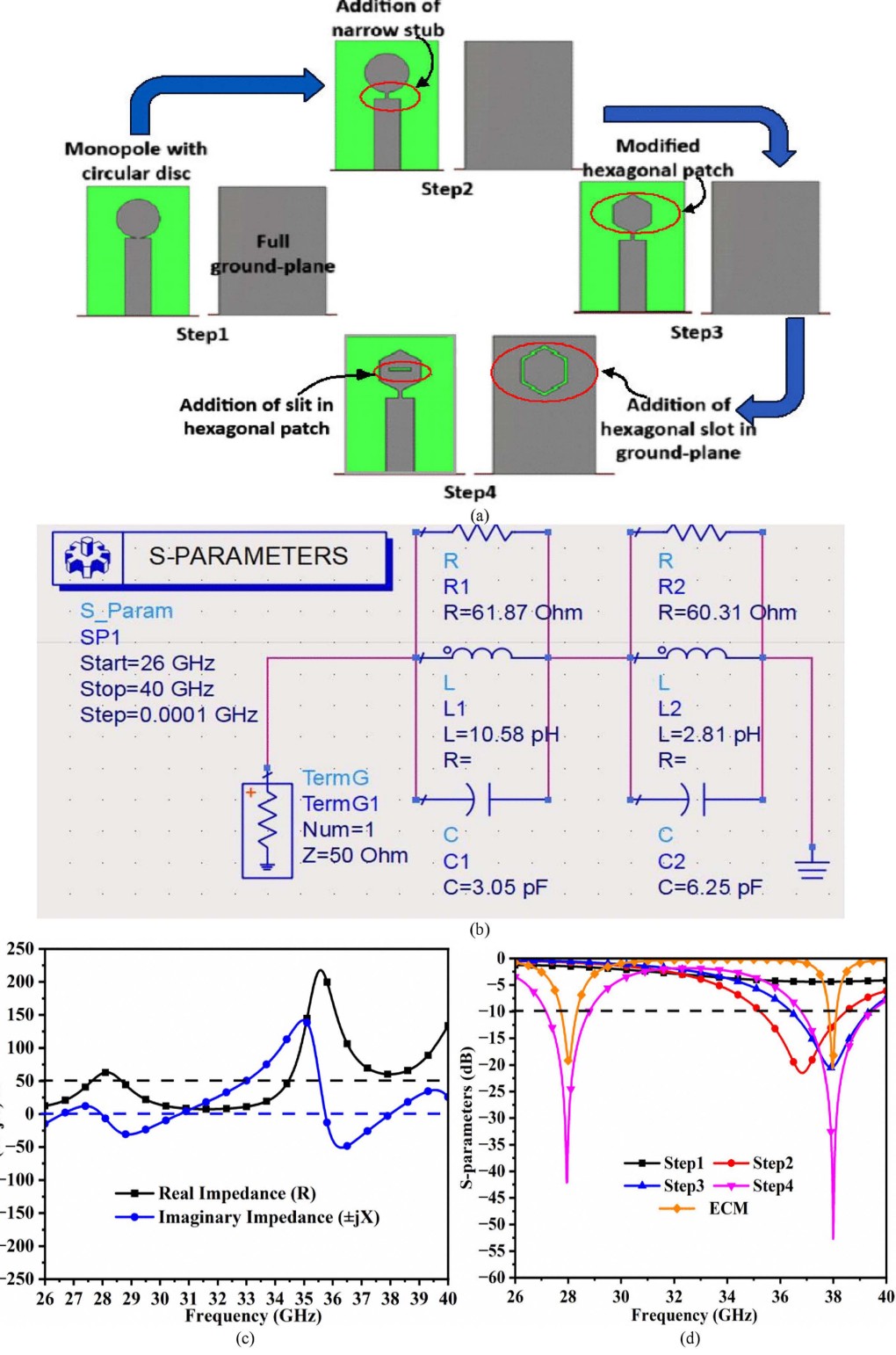

**Fig 2. Evolution and equivalent-circuit-model (ECM) analysis of single-port mmWave antenna.** (a) Evolution steps (b) ECM (c) Re-Img. Graph (d) S-parameters (Evolution and comparison).

$$L = \frac{Img(Z_{11})}{2\pi f_0}$$

(6)

$$C = \frac{1}{(2\pi f_0)^2 L}$$

(7)

The resistive part, which is real impedance and imaginary values are noted from Fig 2(c), with resonance at 28.0 GHz, corresponds to a net impedance of (61.87-j5.97) Ω, and at 38.0 GHz, the total impedance is (60.31 + j0.67) Ω. The bandwidth comparison generated from the EM-simulator and extracted from ECM is tabulated in Table 2.

Table 2 records the resonances near 28.0 GHz and 38.0 GHz, which are generated by the EM-simulator and extracted from ECM analysis. This also verifies that the single-port antenna is suitable for both millimeter-wave band applications.

The resonance frequency achieved at 28.0 GHz and 38.0 GHz is due to the etching of a hexagonal-ring slot in a full-ground and rectangular-slit in the hexagonal radiating-patch, which is represented in Fig 2 with real and imaginary impedance values approaching the impedance of the above two resonating frequencies. The hexagonal ring, which is etched in the ground, affects the shifting of the resonance value at 28.0 GHz. Fig 3(a) shows the equivalent concentric circles encircling the hexagonal ring with radii corresponding to $R_1$ mm and $R_2$ mm, respectively. The change in both the values simultaneously also changes the resonance value, which is plotted in Fig 3(b). For the values of $R_1$ = 1.60 mm, $R_2$ = 1.40 mm, the resonance value corresponds to 28.646 GHz with $S_{11}$ = −32.23 dB. Also, for the values of $R_1$ = 2.00 mm, $R_2$ = 1.80 mm, the resonance value shifts towards the lower side, irrespective of 28.0 GHz, with a value corresponding to 27.31 GHz with $S_{11}$ = −29.33 dB. The optimal value of $R_1$ = 1.80 mm, $R_2$ = 1.60 mm achieves the required resonance at 27.946 GHz with $S_{11}$ = −42.13 dB. Also, during the variation of the $R_1$, $R_2$ parameters, there is a nominal change in resonance of 38.0 GHz, indicating that the parameter $W_3$ is independent. Fig 3(c) shows the variation of the rectangular-slit $W_3$ etched on the radiating patch, which generates the 38.0 GHz millimeter wave band. The value of resonance for $W_3$ = 1.45 mm and $W_3$ = 1.55 mm corresponds to 38.138 GHz ($S_{11}$ = −35.52 dB) & 37.844 GHz ($S_{11}$ = −34.22 dB). However, for $W_3$ = 1.50 mm, exact resonance at 37.984 GHz is achieved with $S_{11}$ = −52.70 dB. Also, the variation of $W_3$ responsible for the 38.0 GHz resonance has no impact on the 28.0 GHz millimeter-wave band.

This parametric analysis concludes that the key parameters $R_1$, $R_2$, and $W_3$ dimension change also change in the resonance value, but are independent of one another.

## 3. The analysis of the Unit Cell frequency-selective-surface (FSS)

The frequency-selective-surface, also known as FSS, is a two-dimensional repetitive periodic structure which functions as a spatial filter (band-stop or band-pass) for plane electro-magnetic waves that are incident on it. The FSS can be a planar structure, which can be either single or multiple stacked periodic layers, which are imprinted either on the top or both surfaces of the dielectric material. The periodic structures utilize geometries such as patches (circular, rectangular-ring, hexagonal-ring), loops, convoluted shapes, and even fractal geometries. The frequency response is produced, which is dependent on the geometry of the structure within the period known as the unit cell. The metallic structure producing

**Table 2. S11 and ECM bandwidth comparison.**

| Bandwidth Extraction | n257 (26.50–29.50 GHz) | | n260 (37.0–40.0GHz) | |
| --- | --- | --- | --- | --- |
| | Bandwidth (GHz) | Resonance (GHz)/$S_{11}$ (dB) | Bandwidth (GHz) | Resonance (GHz)/$S_{11}$ (dB) |
| EM-simulator | 27.162-27.786 | 27.946/-42.13 | 36.794-39.426 | 37.994/-52.70 |
| ECM | 27.7334-28.33 | 28.03/-19.49 | 37.8135-38.11 | 37.9631/-20.64 |

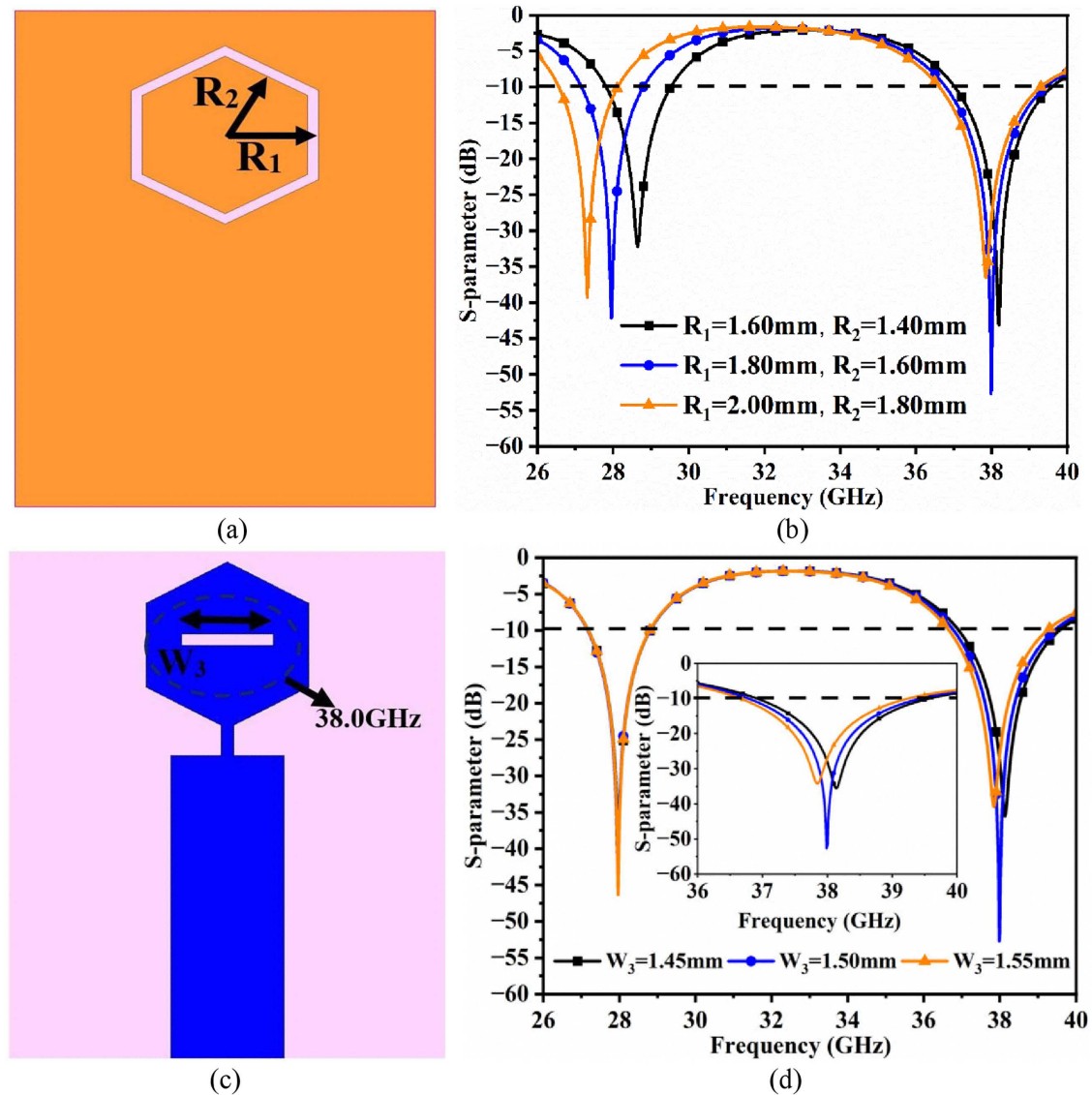

**Fig 3. The parametric analysis of key parameters affecting resonance frequency.** (a) R1, R2 (b) W3.

frequency-response irrespective of filtering characteristics can work as either low-pass, high-pass, band-pass, or even band-stop filters. Also, the FSS includes an important characteristic such as a reduction in side-lobe level radiations and thus focuses on the main-lobe radiation transmission. However, in microwave and millimeter-wave applications, the reflection control is very important because it reduces the interference and also signal degradation. The uncontrolled reflection leads to multiple-path effects where the identical signal is transmitted, reaching the receiver by different paths, leading to destructive interference, which reduces the quality of the signal.

Fig 4 illustrates the design of the single-unit FSS cell reflecting the signals at 28.0 GHz and 38.0 GHz. Fig 4(a) shows the 3D model of the FSS where the metal-reflector is printed on the top-surface of the Rogers5880 substrate with FSS dimensions of $FSS_x \times FSS_y = 3.75$ mm $\times 3.75$ mm and thickness of ha $= 0.787$ mm. Fig 4(b) shows the front-view of the FSS with two concentric hexagonal metallic rings of thickness 0.035 mm printed on the top surface. The outer ring of thickness

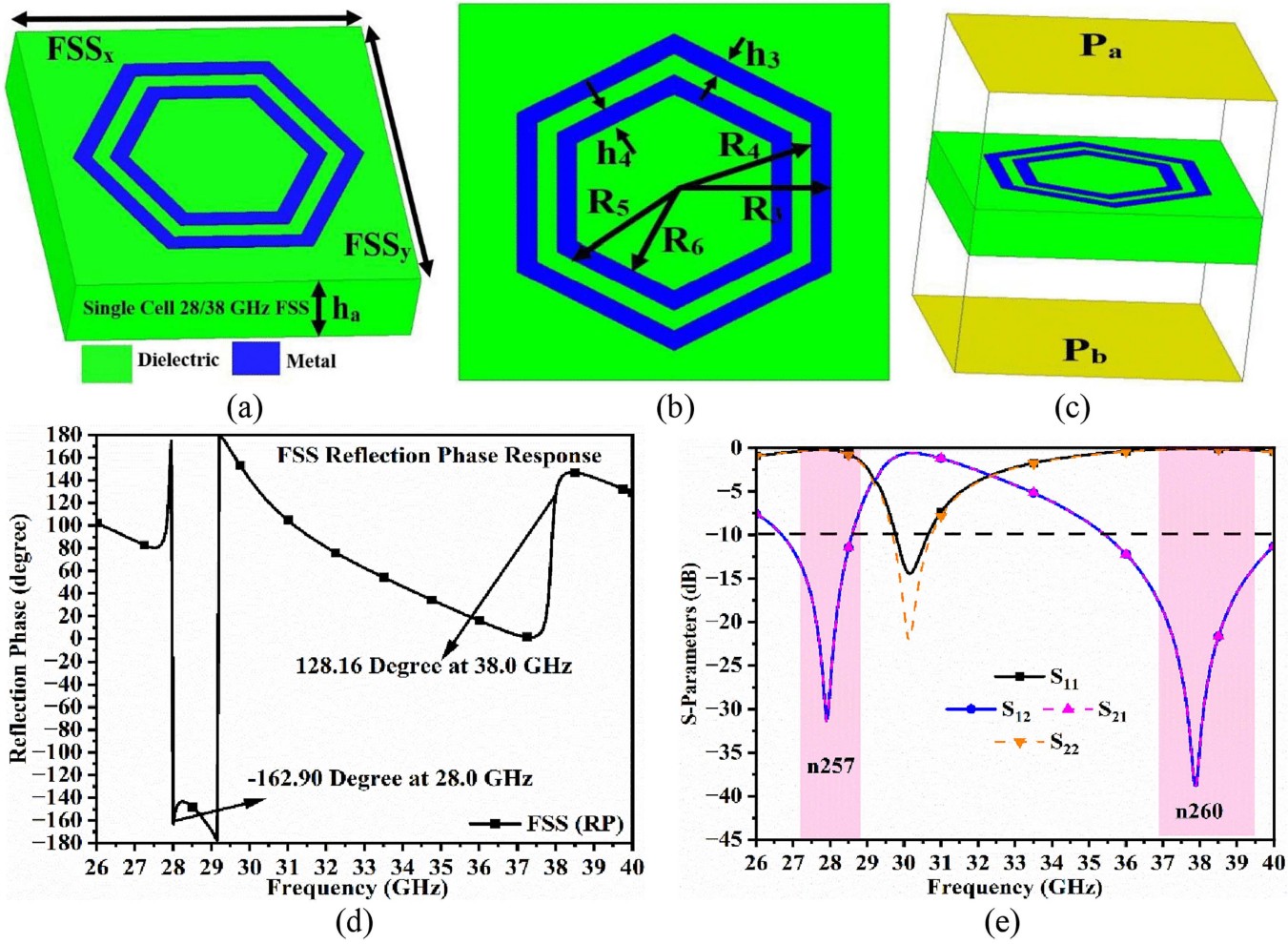

**Fig 4. The FSS unit-cell. (a)** 3D-view **(b)** Optimal-dimensions **(c)** Simulation-model **(d)** Reflection phase response **(e)** S-parameter analysis.

$h_3 = 0.20$ mm is formed by a hexagonal geometry with an equivalent circle of radius $R_3 = 1.58$ mm and $R_4 = 1.38$ mm, which is responsible for stopping the 28.0 GHz millimeter-wave band. Similarly, the inner hexagonal ring of radius $R_5 = 1.18$ mm and $R_6 = 0.98$ mm with thickness of $h4 = 0.20$ mm is designed to stop the 38.0 GHz millimeter-wave band. Fig 4(c) shows the simulation model of the FSS placed within the closed boundary with side-walls $E_t = H_t = 0$ electric and magnetic boundary conditions. The two ports, Pa above the metal-patch and Pb below the opposite face, as shown in Fig 4(c), are used to study the performance of the FSS unit-cell in terms of S-parameters as shown in Fig 4(e). The reflection coefficients ($S_{11}/S_{22}$) graze the 0.0 dB in both the millimeter-wave bands (n257/n260), indicating the signals are reflected from the surface. Also, the transmission coefficients ($S_{12}/S_{21}$) indicate that very minimal power, about 3% is propagated from Porta to Portb. Also, Fig 4(d) shows the reflection phase response versus frequency where the phase angle of −162.90 at 28.0 GHz and 128.16 degree at 38.0 GHz is recorded.

## 4. Two-port dual-band MIMO antenna

The modern wireless communication demands a higher data rate of transmission, improved reliability, and enhanced spectral efficiency. Also, the multiplexing technique and spatial diversity are exploited by MIMO-configuration using

multiple transmitters and receivers. The MIMO antenna system also enhances the channel capacity, which is achieved by transmitting multiple digital signals over the same frequency band.

Fig 5 illustrates the two-port MIMO antenna, which is the extension of the single-port antenna discussed in Fig 1. The 3D-view of the two-port millimeter-wave antenna generating resonance at 28.0 GHz and 38.0 GHz is shown in Fig 5(a), where the two-radiating patches are placed adjacent to each other and separated by $S_a = 10.0$ mm. The new dimension of the two-port antenna corresponds to $L_a \times W_{DP} = 10 \times 17$ mm$^2$. Also, both radiating patches, $A_a$ and $A_b$, share the common-ground excited by RF-connectors (RFC), $RFC_a$ and $RFC_b$. Fig 5(b) shows the S-parameter results where the first antenna, Aa, generates an operating bandwidth of 27.148 GHz-28.80 GHz with a resonance frequency of 27.96 GHz ($S_{11} = -33.123$ dB) and 33.864 GHz-39.51 GHz with a resonance centered at 38.012 GHz ($S_{11} = -52.04$ dB). Similarly, the adjacent antenna, $A_b$, also generates the identical 28.0 GHz and 38.0 GHz bandwidth overlapping with the bandwidth generated by $A_a$. The overlapping transmission coefficients or isolation between antenna $A_a$ and antenna $A_b$ is shown in Fig 5(b) with isolation of more than 26.0 dB in the n257 band and improved isolation of more than 35.0 dB in the n260 band. The 3D-radiation patterns at 28.0 GHz and 38.0 GHz are also shown in Fig 5(c) and Fig 5(d), with wide-spread radiation patterns achieving maximum peak-gain of 4.80 dBi at 28.0 GHz and 7.37 dBi at 38.0 GHz. Fig 5(e) and Fig 5(f) show the surface-current-density (SCD) simulation at 28.0 GHz and 38.0 GHz with antenna $A_b$ excited by an input signal & antenna $A_a$ terminated by a 50 Ω impedance. In both cases, QWT shows the good matching between the transmission-line and the radiating-patch where maximum SCD is concentrated within the QWT, and the radiating-patch observes minimal SCD as the signals are radiated for n257 & n260 bands.

## 5. Four-port mmWave MIMO antenna loaded with dual-narrow-band frequency-selective-surface (FSS) and SAR analysis

The faster data rate with enhanced reliability and reduced multi-path effects is achieved by increasing the radiating elements from two to four in numbers. The four-port version of the dual-band millimeter-wave antenna is shown in Fig 6. The 3D-view of the four-port configuration is shown in Fig 6(a) with the dimensions of $W_{FP} \times L_{FP} = 17.0 \times 22.0$ mm$^2$. The four-radiating patch are identified as $A_1$, $A_2$, $A_3$, $A_4$ and respective connectors are marked as $P_1$, $P_2$, $P_3$, $P_4$. The antennas, $A_1$ and $A_2$, are adjacent to one another as shown in Fig 6(a), while the remaining two antennas, $A_3$ and $A_4$, are oriented by 180°. The micromachining either of the substrate or selectively depositing material is done for microwave, millimeter or even at Terahertz range for controlling the impedance matching, radiation efficiency and also the tailoring of the bandwidth can be controlled. The micromachining includes the following features

(a) Miniaturization: The RF integration is effectively achieved where micromachining enables compact antenna design

(b) Improved radiation efficiency: Removal of the dielectric generates air cavities which reduces dielectric losses

(c) Frequency-stability: The low parasitic effects at high frequency facilitates applications in mmWave and THz range

(d) The micromachining antennas are well suited for wireless communication systems, satellite and space systems, automotive RADAR and biomedical systems

Also, the dielectric substrate is micromachined so that the isolation can be improved by reducing the propagation of surface waves. Fig 6(b) shows the optimal dimension marked in the front view. The inter-spacing between the adjacent radiating elements is $D_2 = 10.0$ mm, and the spacing between the 180° oriented patch is $D_1 = 4.00$ mm. The dielectric substrate is also micromachined with an area of $W_{mm} \times L_{mm} \times h_a = 2.0$ mm $\times 6.0$ mm $\times 0.787$ mm, which is shown in Fig 6(b). The fabricated prototypes are shown in Fig 6(c), Fig 6(d) and Fig 6(e) where the photo-lithographic method of fabrication achieves high precision in dimension of the antenna with accuracy etching of slits and slots in the radiating-patch and ground. Fig 6(f) and Fig 6(g) show the surface-current-density (SCD) simulation with single-port and four-port excitation. Fig 6(f) shows that antenna $A_1$ is excited, and the remaining antennas $A_2$, $A_3$, and $A_4$ are terminated by a matched

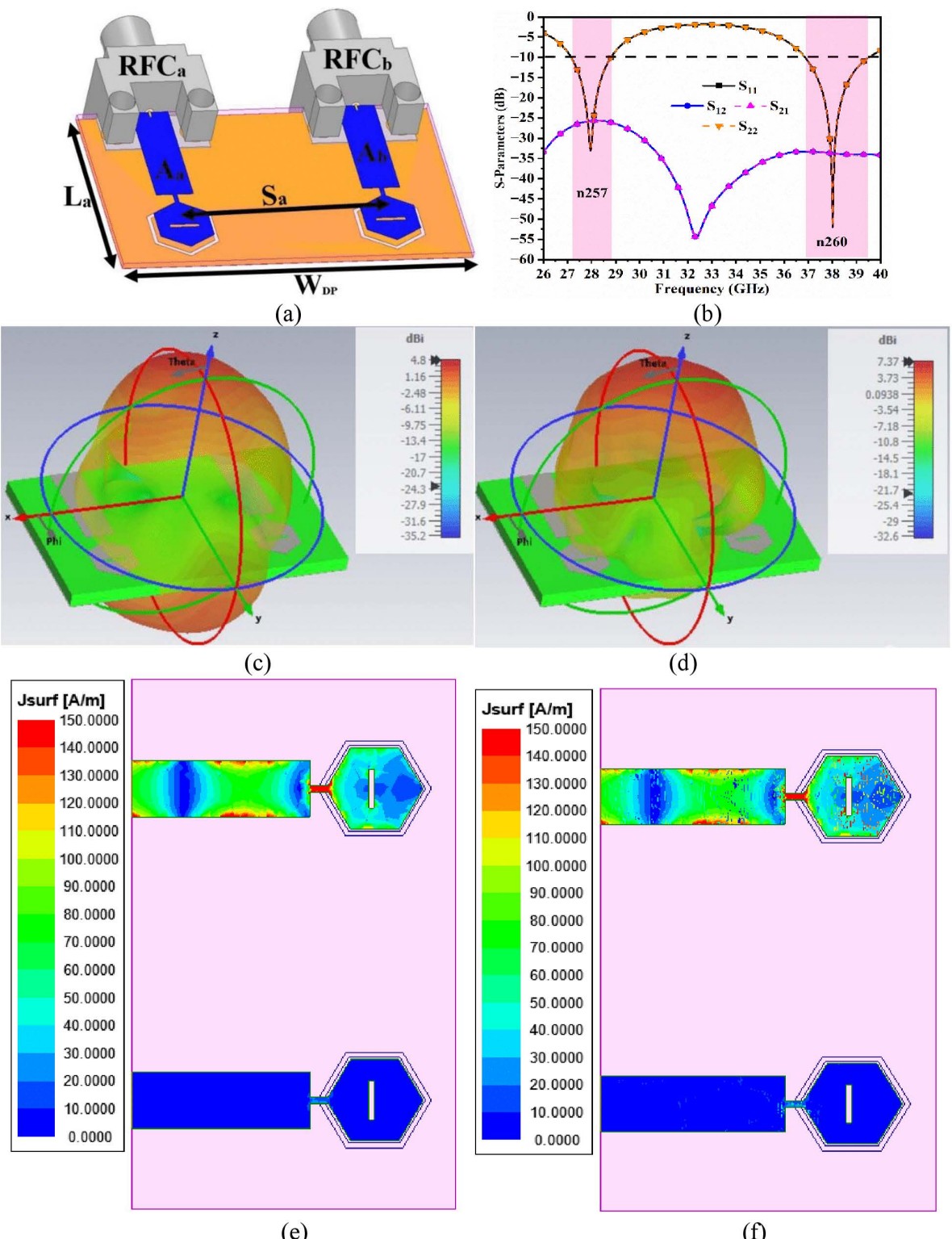

**Fig 5. The dual-band MIMO antenna.** (a) Patch-arrangement in 3D-view (b) S-parameter(s); 3D-pattern at (c) 28.0 GHz (d) 38.0 GHz; SFD analysis at (e) 28.0 GHz (f) 38.0 GHz.

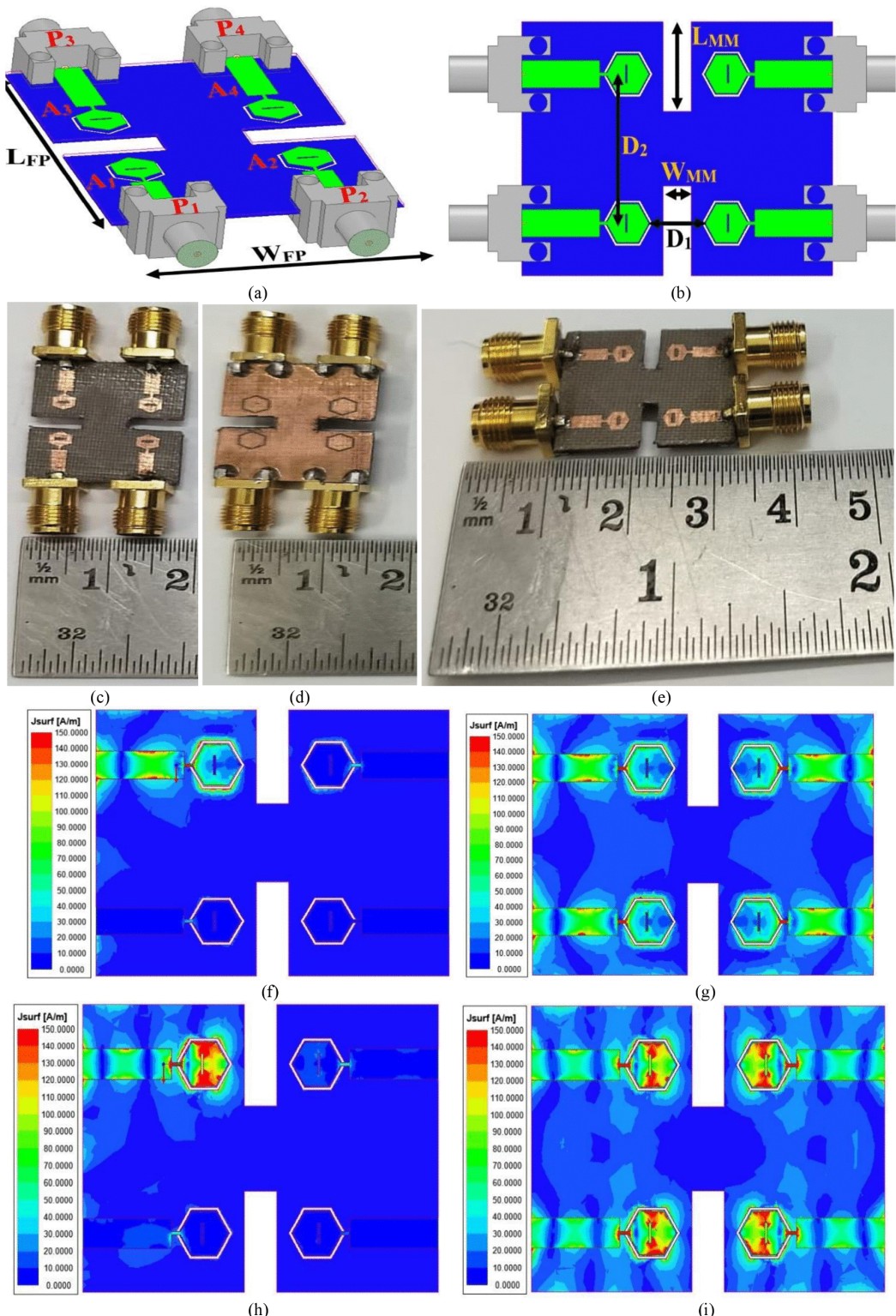

**Fig 6. The dual-band four-port MIMO antenna.** (a) Isometric-view (b) Optimal-dimension; SFD at (f)-(g) 28.0 GHz with single-port and four-port excitation (h)-(i) 38.0 GHz with single-port and four-port excitation.

impedance of 50 Ω. The single-port excited antenna shows that the radiating patch does not store any energy and also offers no coupling to the neighboring antennas $A_2$, $A_3$, and $A_4$. Also, Fig 6(g) shows all four-port antennas excited at the n257 band (28.0 GHz) and concludes that all the antennas radiate energy effectively with no coupling between the radiating elements. Also, Fig 6(h) and Fig 6(i) show the SCD analysis at 38.0 GHz, which also confirms that the antenna radiates effectively with minimal interference.

The unit-cell FSS discussed in Fig 4 showed the capability of acting as a band-stop filter for n257 and n260 millimeter bands. This indicates that all the signals for above said bands are reflected, and this characteristic of FSS can be used with the antenna to enhance the gain. This is achieved by placing an FSS-array of 11 × 11 below the four-port millimeter-wave antenna, where the back-lobe is redirected towards the main lobe and hence increases the gain of the antenna. Also, the unit cell characterizes the behavior, but its array configuration realizes the functionality. Fig 7 gives the details of the FSS-array, its loading with a MIMO four-port millimeter-wave antenna, and the comparison of simulated and measured reflection and transmission coefficients.

Fig 7(a) illustrates an 11 × 11 printing of the FSS unit-cell in two dimensions with an overall size of $FSS_{xx} × FSS_{yy} = 41.25$ mm × 41.25 mm printed on Rogers5880 dielectric material with a thickness of 0.787 mm. Fig 7(b) shows the fabricated prototype of the 121-element FSS-array, which functions as a band-stop filter for n257 and n260 millimeter-wave bands. Fig 7(c) and Fig 7(d) show the simulated result of the 2D-pattern at 28.0 GHz and 38.0 GHz, where the plane wave is incident on the surface of the FSS-array. Fig 7(c) shows the plot of 2D-radiation pattern at 28.0 GHz with main lobe magnitude of 41.90 dB with directed main lobe in 0° and 180° direction. Also, the beam-width at 3.0 dB corresponds to 13°,which is very narrow with a minimal side-lobe level of −12.70 dB. Also, Fig 7(d) shows the 2D-pattern at 38.0 GHz with a main lobe magnitude of 57.70 dB with a more reduced side-lobe level of −13.10 dB with an angular-width of 9.7°. The 3D-view integration in simulation and fabricated prototype (FSS-array plus MIMO antenna) is shown in Fig 7(e) and Fig 7(f), where the gap of d is calculated as

$$d = \frac{\lambda}{2} \; (\lambda = 26.0 \; GHz) \cong \frac{11.50 \; mm}{2} = 5.75 \; mm \tag{8}$$

where **d** indicates the minimal distance between the MIMO antenna and the FSS-array. For accurate dimensions of the fabricated prototype (Antenna and FSS-array) shown in Fig 7(f), the photo-lithographic method of fabrication is used.

Fig 7(g) and Fig 7(h) illustrate the 3D-radiation pattern supporting the function of FSS as a reflector at 28.0 GHz and 38.0 GHz. The simulated-measured reflection coefficients with the MIMO antenna loaded with FSS are shown in Fig 7(i) and Fig 7(j). The simulated −10.0 dB bandwidth for antenna $A_1$ corresponds to 27.246 GHz-28.84 GHz with resonance centered at 28.03 GHz ($S_{11} = −42.22$ dB). Also, due to the symmetrical structure, the remaining antennas $A_2$, $A_3$, and $A_4$ also produce the results. Also, the MIMO antenna covers the n260 band with −10.0 dB bandwidth of 36.864 GHz-39.412 GHz with resonance frequency centered at 37.984 GHz ($S_{11} = −43.50$ dB). Fig 7(j) illustrates the plot of measures −10.0 dB bandwidth of 26.45 GHz-29.27 GHz (n257) and 37.04 GHz-39.12 GHz (n260) respectively. The transmission coefficients or isolation are plotted in Fig 7(k) and Fig 7(l) with simulation records showing isolation of more than 22.50 dB and measurement isolation records of more than 19.18 dB.

The $MIMO_{mmWave-FSS}$ antenna, which includes four radiating elements $A_1$, $A_2$, $A_3$, and $A_4$, as shown in Fig 6(a), radiates the electromagnetic signals independently. However, their orientation ensures not only the inter-spaced isolation but also maintains the desired operational bandwidth. Hence, the correlation between the radiation patterns generated by individual antennae is measured by the Envelope-Correlation-Coefficient ($ECC_{mmWave-FSS}$).

$$\gamma_{c(mmWave-FSS)} = \frac{\int_0^{2\pi} \int_0^{\pi} \left( (XPRE_{\theta.m}(\theta, \phi) \, E^*_{\theta,s}(\theta, \phi) P_\theta(\theta, \phi) + \; E_{\phi.m}(\theta, \phi) \, E^*_{\phi,s}(\theta, \phi) P_\phi(\theta, \phi) \right) sin\theta \; d\theta \; d\phi}{\sqrt{\delta_m^2} \sqrt{\delta_s^2}} \tag{9}$$

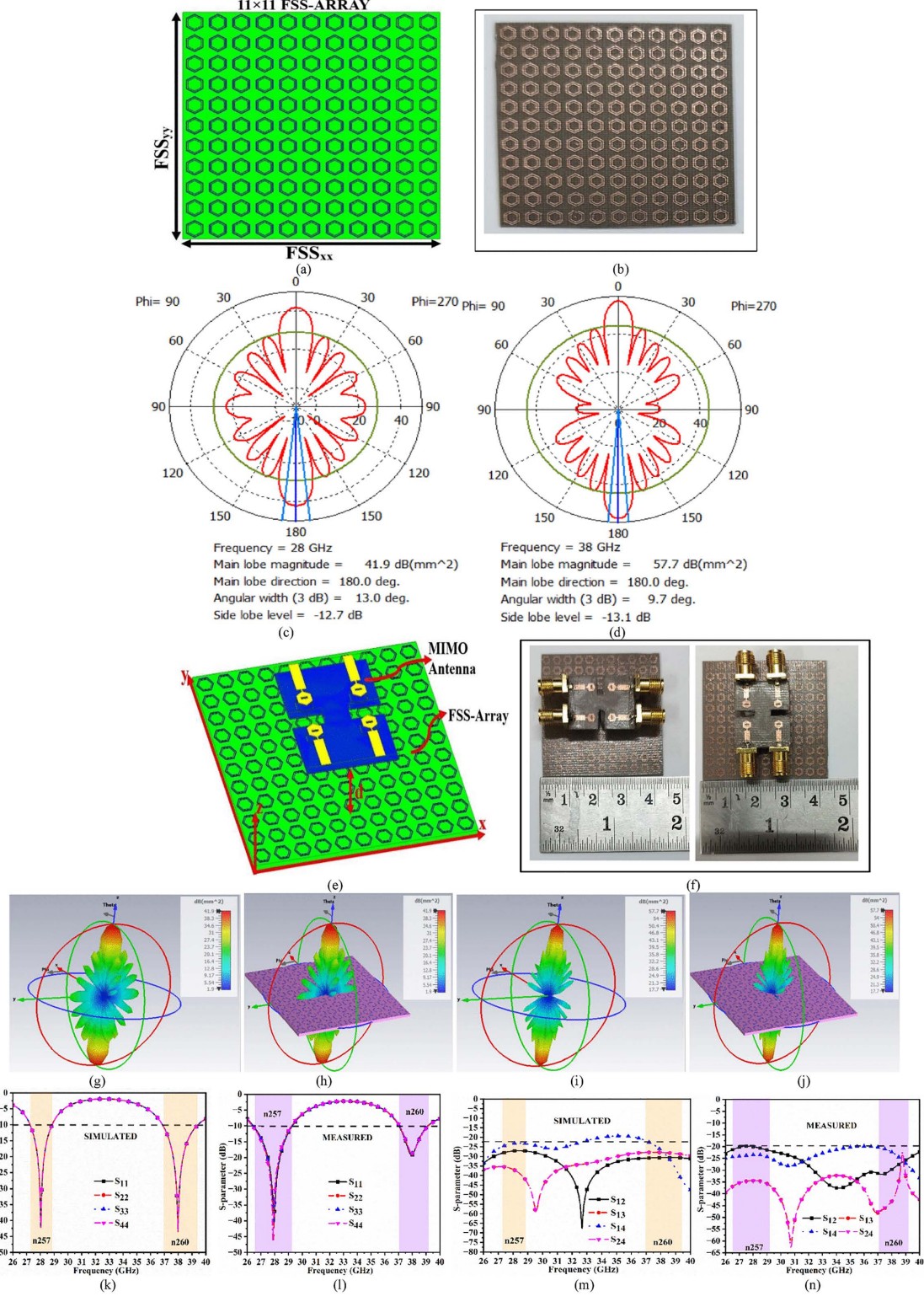

**Fig 7. Loading of FSS-array with MIMO antenna. (a)** Simulation model of FSS **(b)** FSS-array prototype **(c)**-(d) 2D-radiation patterns at 28.0 GHz and 38.0 GHz **(e)**-(f) 3D-view of simulated-fabricated MIMO antenna loaded with FSS **(g)**-(h) 3D-radiation patterns at 28.0 GHz without and 38.0 GHz **(i)**-(j) Simulated-Measured reflection-coefficients **(k)**-(l) Simulated-Measured transmission-coefficients.

Here, $\delta_m^2$ and $\delta_s^2$ Is the variance related to ports, which can be further mathematically expressed as

The ECC$_{\text{mmWave-FSS}}$ helps in quantifying the degree of correlation between the signal received by each of the radiating elements by the receiver. The more similar the radiation patterns, the better the diversity and the higher the data throughput. The value of ECC$_{\text{mmWave-FSS}}$ lies between 0 and 1, with 0 indicating the exact replication of radiation patterns by each of the radiating-element while 1 indicates the highly deteriorated radiation patterns due to maximum interference between them. The values of ECC$_{\text{mmWave-FSS}}$ for the operational bandwidth must be ideally less than 0.50, which are calculated by Equation (9) using the 3D-radiation method. Table 3 records the simulated and measured ECC$_{\text{mmWave-FSS}}$ values in n257 and n260 millimeter-wave bands noted from Fig 8(a) and Fig 8(b). The simulated-measured values are less than 0.18 & 0.132, respectively, for the n257 band and less than 0.175 & 0.225 in the n260 band, which are less than the ideal value (<0.50).

$$DG_{\text{M-FSS}} = 10\sqrt{1 - |\rho_e|^2}$$

(10)

The Diversity-Gain (DG$_{\text{mmWave-FSS}}$) measures the signal reliability of the MIMO-radiating elements. Fig 8(c) and Fig 8(d) record the simulated-measured values of DG$_{\text{mmWave-FSS}}$ in n257 and n260, which are calculated by Equation (10). The DG$_{\text{mmWave-FSS}}$ also signifies the mitigation of fading effects, thereby evaluating the merit of improvement in signal quality or error-reduction rate, and the values of DG$_{\text{mmWave-FSS}}$ ideally must be more than 9.95 dB. Both bands record the value more than 9.995 dB, as noted in Table 3.

$$\Gamma_a^t = \frac{\text{Available Power (AP)} - \text{Radiated Power (RP)}}{\text{Available Power (AP)}}$$

(11)

$$\Gamma_a^t = \frac{\sqrt{\sum_{i=1}^N |b_i|^2}}{\sqrt{\sum_{i=1}^N |a_i|^2}}$$

(12)

$$b_1 = S_{11}a_1 + S_{12}a_2 = S_{11}a_0e^{j\theta_1} + S_{12}a_0e^{j\theta_2} = a_1\left(S_{11} + S_{12}e^{j\theta}\right)$$

(13)

$$b_2 = S_{21}a_1 + S_{22}a_2 = S_{21}a_0e^{j\theta_1} + S_{22}a_0e^{j\theta_2} = a_1\left(S_{21} + S_{22}e^{j\theta}\right)$$

(15)

$$\Gamma_a^t = \frac{\sqrt{\left|S_{11} + S_{12}e^{j\theta}\right|^2 + \left|S_{21} + S_{22}e^{j\theta}\right|^2}}{\sqrt{2}}$$

(14)

The Total-Active-Reflection-Coefficient (TARC$_{\text{mmWave-FSS}}$) indicates the overall return loss of the entire four-port MIMO antenna. TARC$_{\text{M-FSS}}$ also measures how much power is reflected when all the MIMO antennas are excited. The individual

Table 3. Comparison of simulated-measured diversity parameters.

| Diversity-parameters | n257 (26.50–29.50 GHz) | | n260 (37.0–40.0GHz) | | Ideal values |
|---|---|---|---|---|---|
| | Simulated | Measured | Simulated | Measured | |
| ECC$_{\text{mmWave-FSS}}$ | <0.18 | <0.175 | <0.132 | <0.225 | <0.50 |
| DG$_{\text{mmWave-FSS}}$ (dB) | >9.998 | >9.969 | >9.982 | >9.955 | <9.95 |
| TARC$_{\text{mmWave-FSS}}$ (dB) | −4.92 | −5.00 | −5.18 | −4.76 | <0.0 |
| CCL$_{\text{mmWave-FSS}}$ (b/s/Hz) | 0.38 | 0.32 | 0.35 | 0.30 | <0.40 |

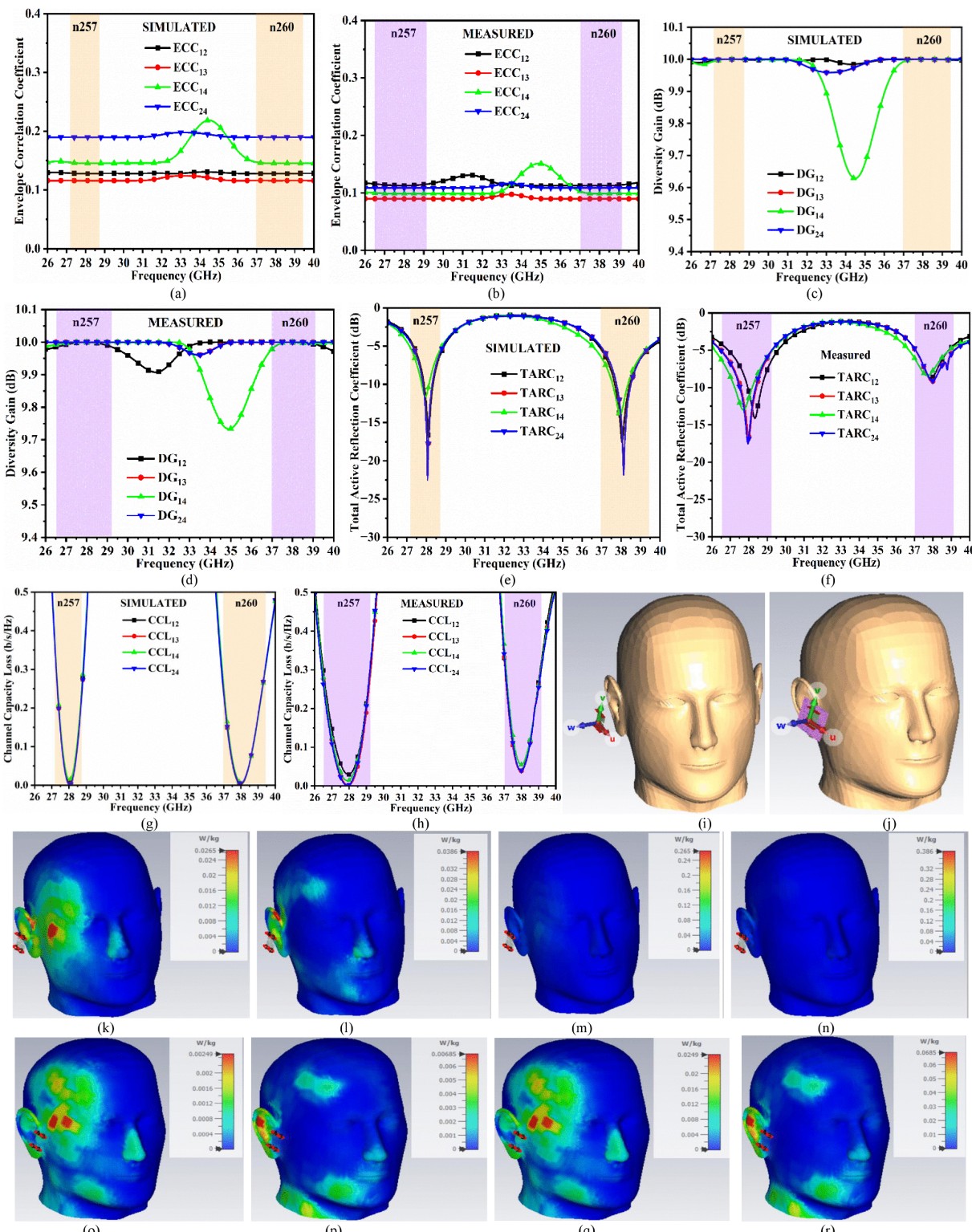

**Fig 8. Simulated-Measured diversity parameters. (a)**-(b) ECC **(c)**-(d) DG **(e)**-(f) TARC **(g)**-(h) CCL; SAR analysis **(i)** MIMO antenna placed near Human-head phantom (MIMO-HHP) model **(j)** MIMO antenna loaded with placed near Human-head phantom (MIMO-HHP) model; SAR analysis ($p_i$ = 50 mW) of MIMO antenna at (k) 28.0 GHz (l) 38.0 GHz; SAR analysis ($p_i$ = 500 mW) of MIMO antenna at (m) 28.0 GHz (n) 38.0 GHz; SAR analysis ($p_i$ = 50 mW) of MIMO antenna loaded with FSS at (o) 28.0 GHz (p) 38.0 GHz; SAR analysis ($p_i$ = 500 mW) of MIMO antenna loaded with FSS at (q) 28.0 GHz (r) 38.0 GHz.

S-parameters ($S_{11}$, $S_{22}$, $S_{33}$, $S_{44}$) correspond to the performance of individual but, but the MIMO system calculates the reflection coefficient cumulatively for all the ports. The TARC$_{mmWave\text{-}FSS}$ is calculated from Equation (11) to Equation (14), and the TARC$_{mmWave\text{-}FSS}$ for the proposed MIMO-FSS antenna is plotted in Fig 8(e) and Fig 8(f). The simulated values are more or less than −4.92 dB in n257 and 5.18dB in n260. Also, the measured values record less than −5.00 dB and −4.76 dB in n257 and n260, respectively.

$$CCL_{MBA} = log_2\left(\det\left(I_{a \times a} + \frac{\rho}{A_t}A\right)\right) - log_2\left(\det\left(I_{a \times a} + \frac{\rho}{A_t}A_{ideal}\right)\right) \tag{15}$$

$$CCL_{MBA} = -log_2 \det\left(\alpha^s\right) \tag{16}$$

where

$$\rho_{mm} = 1 - \sum_{n=1}^{4}\left|S_{mn}\right|^2 \tag{17}$$

$$\rho_{ms} = -\left(S_{mm}^*S_{ms} + S_{sm}^*S_{ms}\right) \tag{18}$$

Channel-Capacity-Loss quantifies the amount of reflection in the data-rate transmission, which is due to the non-idealities in the MIMO antenna system. The CCL$_{mmWave\text{-}FSS}$ depends on the perfect achievement of spatial diversity with higher isolation. Equations (15) to Equation (18) and the corresponding simulated-measured values of CCL are plotted in Fig 8(g) to Fig 8(h). The simulated and measured CCL$_{mmWave\text{-}FSS}$ are less than 0.40 b/s/Hz in n257 and n260. The simulated CCL$_{mmWave\text{-}FSS}$ in n257 and n260 are less than 0.35 b/s/Hz and 0.38 b/s/Hz, respectively. Also, the measured CCL$_{mmWave\text{-}FSS}$ corresponds to less than 0.32 b/s/Hz and 0.30 b/s/Hz in both millimeter-wave bands, Fig 8(g) and Fig 8(h).

$$SAR = \frac{\sigma|E|^2}{\rho} \tag{19}$$

where $\sigma$ is the conductivity of the body tissue (S/m), $E$ is the applied electric field (V/m), and $\rho$ is the mass density of the body tissue (Kg/m³).

Fig 8 also includes specific-absorption-rate (SAR) analysis of the proposed millimeter-wave MIMO antenna loaded with FSS for power input of 50 mW and 500 mW at key resonance frequency values of 28.0 GHz and 38.0 GHz respectively. The study includes tissue-model of the human-head-phantom (HHP) with electrical properties tabulated in Table 4 at resonance frequency values of 28.0 GHz and 38.0 GHz. Fig 8(i) and Fig 8(j) shows the placing of the MIMO antenna without and with FSS near the HHP model at a distance of 15.0 mm. Fig 8(k) to Fig 8(r) illustrates the SAR analysis without & with

**Table 4. Electrical properties of tissue model at various frequencies [43,44].**

| Tissue | Frequency (GHz) | Electrical Permittivity | Conductivity | Loss Tangent | Density (Kg/m³) |
|---|---|---|---|---|---|
| Skin | 28.0 | 16.5 | 25.8 | 1.0016 | 1109 |
| Fat | | 6.09 | 5.06 | 0.29471 | 911 |
| Muscle | | 24.4 | 33.6 | 0.88284 | 1090 |
| Skin | 38.0 | 12.297 | 31.043 | 1.1941 | 1109 |
| Fat | | 3.444 | 2.1358 | 0.29331 | 911 |
| Muscle | | 19.056 | 41.823 | 1.0382 | 1090 |

FSS with power input of 50 mW & 500 mW at frequency 28.0 GHz and 38.0 GHz. The SAR analysis is recorded in Table 5 which shows SAR calculations calculated from Equation (19) where SAR depends on conductivity of the tissue, applied electric-field and mass density of the tissue. The input power of 50 mW records the SAR values of 0.0265 W/kg at 28.0 GHz and 0.0386 W/kg at 38.0 GHz. However, the SAR values are significantly reduced for input power of 50 mW in the presence of FSS loading with MIMO antenna. Identical behavior is also recorded for input power of 500 mW as concluded from Table 5.

## 6. Far-field analysis of Four-port mmWave MIMO antenna loaded with dual-narrow-band frequency-selective-surface (FSS), and state-of-the-art comparison

Fig 9 illustrates the far-field analysis where 2D-radiation patterns, peak-realized gain, and radiation efficiency are discussed. Fig 9(a) and Fig 9(b) show the plot of simulated-measured 2D-radiation patterns at 28.0 GHz and 38.0 GHz with maximum radiation directed in the boresight direction at 180°. The simulated and measured 2D-radiation patterns show that the back-lobe levels are reduced due to the presence of FSS, which reflects the signals at 28.0 GHz and 38.0 GHz. Fig 9(c) shows the simulated-measured radiation efficiency at 28.0 GHz and 38.0 GHz, with simulation values are 86% at 28.0 GHz, 88% at 38.0 GHz, and the measured values correspond to 83% at 28.0 GHz, 91% at 38.0 GHz, respectively. Fig 9(d) also includes the simulated-measured peak-realized-gain noted at 28.0 GHz and 38.0 GHz, respectively. The peak-realized-gain without FSS corresponds to 4.80 dBi at 28.0 GHz and 7.37 dBi at 38.0 GHz, while the measured values at 28.0 GHZ are 9.96 dBi, and at 38.0 GHz the value is 11.48 dBi. The peak-gain rise by 5.16 dBi is noted at 28.0 GHz and 4.11 dBi at 38.0 GHz, which is due to the loading of the FSS-array with a four-port MIMO antenna.

The proposed four-port MIMO antenna loaded with FSS is compared with the earlier published work, as shown in Table 4. The single/dual-band MIMO antenna is not integrated with FSS compared with the proposed work. However, the peak-gain is enhanced by 2.60 dBi [14], and the meta-surface integrated with a millimeter-wave antenna [8] is used to convert linear to circular polarization, but does not enhance the peak-gain. However, the proposed antenna records the enhancement of peak-gain by 5.16 dBi when FSS is loaded, as recorded from Table 6.

## 6. Conclusions

In this current study, a four-port millimeter-wave MIMO antenna loaded with a novel frequency-selective surface (FSS) is investigated. The four-port MIMO antenna is printed on Rogers substrate with an overall area of 374 mm$^2$, generating two narrow bands in n257 (28.0 GHz) and n260 (38.0 GHz). Also, the MIMO antenna is loaded with a novel FSS-array which performs the role of gain-enhancement with a size of 42.50 mm × 42.50 mm × 0.787 mm printed on Rogers substrate. The MIMO antenna also features a lower value of ECC$_{mmWave-FSS}$, high DG$_{mmWave-FSS}$, CCL$_{mmWave-FSS}$ less than 0.30, and TARC$_{mmWave-FSS}$ less than −4.76 dB. The MIMO antenna also achieves a maximum peak-realized-gain of 9.96 dBi at 28.0 GHz, 11.48 dBi at 38% with radiation efficiency of more than 83% and highly directed 2D-radiation patterns with suppressed back-lobe radiations. The SAR values of MIMO antenna loaded with FSS are also recorded with values 0.00249 W/kg at 28.0 GHz, 0.00685 W/kg at 38.0 GHz (power input = 50 mW) and 0.0249 W/kg at 28.0 GHz, 0.0685 W/kg at 38.0 GHz (power input = 500 mW).

**Table 5. SAR values of eight-port MIMO antenna without and with FSS.**

| Frequency (GHz) | SAR (W/kg) Without FSS | SAR (W/kg) With FSS | Power Input (mW) |
|---|---|---|---|
| 28.0 | 0.0265 | 0.00249 | 50 |
| | 0.265 | 0.0249 | 500 |
| 38.0 | 0.0386 | 0.00685 | 50 |
| | 0.386 | 0.0685 | 500 |

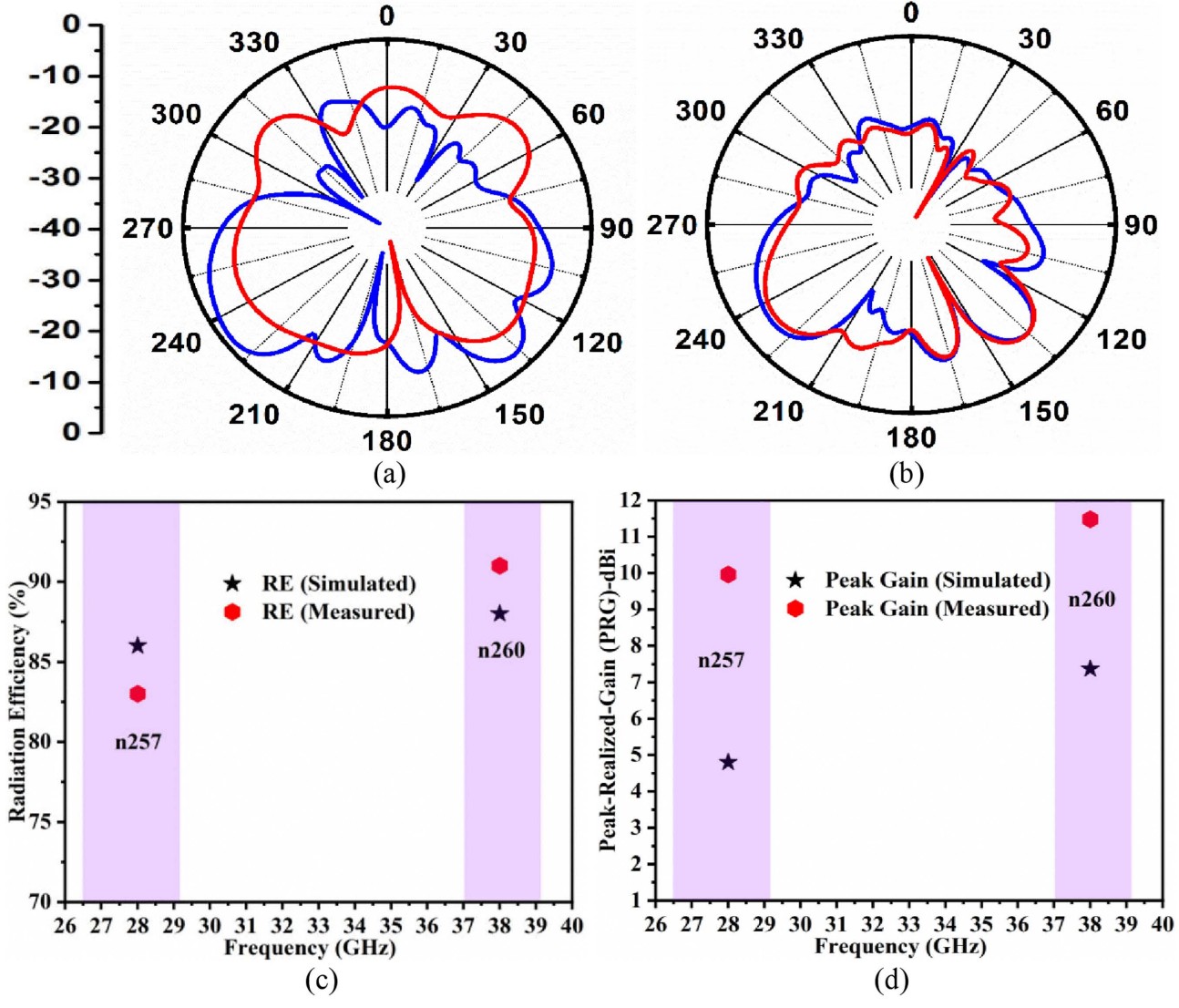

**Fig 9. Simulated-Measured far-field results. (a)**-(b) 2D-radiation patterns at 28.0 GHz and 38.0 GHz **(c)**-(d) Radiation efficiency in n257 and n260 millimeter-wave bands.

## Author contributions

**Conceptualization:** Manish Sharma.

**Data curation:** Geetanjali Singla.

**Formal analysis:** Bhaskara Rao Perli.

**Software:** Tathababu Addepalli.

**Validation:** B. Satya Sridevi.

**Visualization:** Sivasubramanyam Medasani.

**Writing – original draft:** Tanweer Ali.

**Writing – review & editing:** Tanweer Ali.

**Table 6. State-of-the-art comparison of the proposed MIMO antenna with FSS.**

| Ref./ Year | Size (mm²) | Bandwidth (GHz) | No. of Ports/ Isolation (dB) | ECC/DG (dB) | TARC (dB)/CCL (b/s/Hz) | Peak-Gain | FSS Integration/Gain Enhancement (dB) |
|---|---|---|---|---|---|---|---|
| [1] 2024 | 20.48 × 20.48 | 25.21-32.34 | 04 >20.0 | <0.006 >9.97 | <−10.0 <0.135 | 6.87 | NO NO |
| [4] 2025 | 34.29 × 34.29 | 25.05-30.15 | 04 >31.37 | <0.0002 >9.996 | <−10.0 <0.30 | 7.10 | NO NO |
| [8] 2024 | 6.0 × 7.50 | 36.0-40.0 | 02 >20.0 | <0.50 >9.99 | NC <0.50 | 7.20 | Yes Used to convert linear to circular polarization |
| [11] 2022 | 30.0 × 30.0 | 27.10-28.0 | 04 >32.0 | <0.0005 >9.999 | NC <0.15 | 7.10 | NO NO |
| [13] 2023 | 48.0 × 12.0 | 37.75-41.0 | 04 >20.0 | <0.00015 >9.999 | NC NC | 4.90 | NO NO |
| [14] 2023 | 25.95 × 25.95 | 37.20-39.20 | 04 >25.0 | <0.005 >9.99 | NC <0.40 | 8.40 | Yes 2.60 |
| [20] 2023 | 60.0 × 60.0 | 27.35-30.40 36.98-39.40 | 04 >30.0 | <0.001 >9.997 | NC <0.15 | 8.14 | NO NO |
| [21] 2023 | 26.0 × 26.0 | 27.70-28.30 37.70-38.30 | 04 >30.0 | <0.0001 >9.99 | NC <0.30 | 8.10 | NO NO |
| [23] 2025 | 24.0 × 24.0 | 27.50-28.40 37.50-39.50 | 04 >20.0 | <0.01 >9.98 | <−12.0 <0.40 | 5.90 | NO NO |
| [25] 2024 | 28.0 × 28.0 | 27.82-29.11 37.22-38.33 | 04 >19.0 | <0.0005 >9.99 | <−10.0 <0.03 | 7.90 | NO NO |
| [15] 2024 | 18.0 × 8.50 | 27.76-28.48 37.69-38.19 | 04 >20.0 | <0.03 >9.75 | <−10.0 <0.15 | 7.73 | NO NO |
| Proposed | 17.0 × 22.0 | 26.45-29.27 37.04-39.12 | 04 >19.18 | <0.18 >9.955 | <−4.76 <0.30 | 11.48 | Yes 5.16 |

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
