## [Decision Letter · Decision Letter 0]

30 Dec 2025

Micromachined mmWave 28.0/38.0 GHz MIMO Antenna Loaded with Novel Frequency Selective Surface for Gain Enhancement and SAR Analysis for Future Wireless Applications

PLOS One

Dear Dr. Ali,

Thank you for submitting your manuscript to PLOS ONE. After careful consideration, we feel that it has merit but does not fully meet PLOS ONE’s publication criteria as it currently stands. Therefore, we invite you to submit a revised version of the manuscript that addresses the points raised during the review process.

We look forward to receiving your revised manuscript.

Kind regards,

Neng Ye

Academic Editor

PLOS One

Journal Requirements:

Additional Editor Comments:

Thanks for your submitting. As mentioned by the reviewers ,there's still some problems needed to be fixed in your paper.I noticed that the main issues are the insufficient introduction to the current research status, as well as the inadequate explanation of technical details and advantages. Please make revisions to meet the publication standards.

Reviewers' comments:

Reviewer's Responses to Questions

**Comments to the Author**

1. Is the manuscript technically sound, and do the data support the conclusions?

Reviewer #1: Yes

Reviewer #2: Partly

2. Has the statistical analysis been performed appropriately and rigorously?

Reviewer #1: Yes

Reviewer #2: Yes

3. Have the authors made all data underlying the findings in their manuscript fully available?

Reviewer #1: Yes

Reviewer #2: Yes

4. Is the manuscript presented in an intelligible fashion and written in standard English?

Reviewer #1: Yes

Reviewer #2: Yes

Reviewer #1: The authors present the work titled "Micromachined mmWave 28.0/38.0 GHz MIMO Antenna Loaded with Frequency Selective Surface for Gain Enhancement and SAR Analysis for Future Wireless Applications." The concept is promising and relevant; however, I have the following suggestions for improvement:

Title Revision: Please remove the word "novel" from the title regarding the FSS, as these unit cells are already well-studied in the literature.

Introduction: The contributions listed in the introduction are currently too long. Please condense this section to a maximum of four key contributions.

Literature Review: The authors should expand the discussion on recently published works regarding mmWave antennas with FSS reflectors. The following articles are recommended to add value to the introduction:

a. https://doi.org/10.1016/j.hspr.2025.08.004

b. https://doi.org/10.1007/s10762-025-01079-z

c. https://doi.org/10.3390/math9243301

Technical Data: Please provide the reflection phase characteristics of the proposed FSS unit cell.

Reviewer #2: This paper proposes a dual-band four-port millimeter-wave MIMO antenna that achieves significant gain enhancement and systematically evaluates key MIMO performance metrics. The manuscript is well structured, the design procedure is clearly presented, and the parametric analysis is relatively comprehensive. However, the following issues should be addressed.

1. In the Introduction, the related literature is mainly classified based on research topics. It is recommended that the authors further compare these works by discussing their respective advantages and limitations, so as to more clearly highlight the novelty and contributions of this paper.

2. In Section 5, the description of the micro-machining process is rather vague. More details regarding the specific fabrication methods and key process parameters should be provided. In addition, please ensure consistency in the terminology, and clarify whether “micro-machined” should be written with or without a hyphen.

3. The SAR analysis is relatively brief. The authors should explicitly specify the input power level used in the SAR simulations and clarify whether the SAR results are normalized. Otherwise, the reported SAR values cannot be fairly compared with those in other studies.

4. This paper designs a high-isolation, high-gain MIMO antenna, which effectively reduces inter-user interference. It is suggested that the authors strengthen the theoretical foundation supporting this conclusion by citing relevant works, such as “Achieving Positive Rate of Covert Communications Covered by Randomly Activated Overt Users” and “Energy Efficiency of Massive Random Access in MIMO Quasi-Static Rayleigh Fading Channels With Finite Blocklength”, to make the claims more convincing.

5. Please pay more attention to writing conventions and consistency. For example, the reference citation format is inconsistent in the Introduction, where references [1] and [3] are used differently. In addition, some abbreviations (e.g., ECC) are not defined when they first appear in the text.

**Do you want your identity to be public for this peer review?** For information about this choice, including consent withdrawal, please see our Privacy Policy

Reviewer #1: No

Reviewer #2: No

---

## [Author Response · Author response to Decision Letter 1]

8 Jan 2026

Additional Editor Comments:

Thanks for your submitting. As mentioned by the reviewers, there's still some problems needed to be fixed in your paper. I noticed that the main issues are the insufficient introduction to the current research status, as well as the inadequate explanation of technical details and advantages. Please make revisions to meet the publication standards.

Authors are thankful to editor for giving the opportunity to revise the manuscript titled “Micromachined mmWave 28.0/38.0 GHz MIMO Antenna Loaded with Novel Frequency Selective Surface for Gain Enhancement and SAR Analysis for Future Wireless Applications,”.

Also, we are giving point-to-point for the queries raised by the reviewer(s).

Reviewer #1: The authors present the work titled "Micromachined mmWave 28.0/38.0 GHz MIMO Antenna Loaded with Frequency Selective Surface for Gain Enhancement and SAR Analysis for Future Wireless Applications." The concept is promising and relevant; however, I have the following suggestions for improvement:

Title Revision: Please remove the word "novel" from the title regarding the FSS, as these unit cells are already well-studied in the literature.

Response: Authors are thankful to reviewer for removal of word “novel”. As per your suggestions, the above said word has been removed from the title of the revised manuscript.

Introduction: The contributions listed in the introduction are currently too long. Please condense this section to a maximum of four key contributions.

Response: Authors have condensed the introduction section as suggested by the reviewer with focusing maximum of four key contributions.

Literature Review: The authors should expand the discussion on recently published works regarding mmWave antennas with FSS reflectors. The following articles are recommended to add value to the introduction:

a. https://doi.org/10.1016/j.hspr.2025.08.004

b. https://doi.org/10.1007/s10762-025-01079-z

c. https://doi.org/10.3390/math9243301

Response: Authors are thankful to reviewer for suggesting the latest state-of-the-art research articles. We have included the above said research articles and also highlighted the same in revised manuscript with updating in introduction.

Technical Data: Please provide the reflection phase characteristics of the proposed FSS unit cell.

Response: Authors are thankful to reviewer for addressing the concern related to reflection phase characteristics. The analysis of reflection phase characteristics versus frequency is added in the revised manuscript as Figure 4(d).

Reviewer #2: This paper proposes a dual-band four-port millimeter-wave MIMO antenna that achieves significant gain enhancement and systematically evaluates key MIMO performance metrics. The manuscript is well structured, the design procedure is clearly presented, and the parametric analysis is relatively comprehensive. However, the following issues should be addressed.

1. In the Introduction, the related literature is mainly classified based on research topics. It is recommended that the authors further compare these works by discussing their respective advantages and limitations, so as to more clearly highlight the novelty and contributions of this paper.

Response: Authors are thankful to reviewer for raising the concerns over introduction. The introduction has been modified as per the suggestions and same has been highlighted in the revised manuscript.

2. In Section 5, the description of the micro-machining process is rather vague. More details regarding the specific fabrication methods and key process parameters should be provided. In addition, please ensure consistency in the terminology, and clarify whether “micro-machined” should be written with or without a hyphen.

Response: Authors are thankful for the description of micromachining antenna discussed in Section 5. The features of the micromachining antenna is explained in the revised manuscript and same has been highlighted. Also, the hypen is removed and “micromchined” word is used in the entire manuscript.

3. The SAR analysis is relatively brief. The authors should explicitly specify the input power level used in the SAR simulations and clarify whether the SAR results are normalized. Otherwise, the reported SAR values cannot be fairly compared with those in other studies.

Response: Authors are again thankful for addressing the concerning the power input in calculation of SAR. In this study of mmWave loaded with FSS, the SAR values at 28.0 GHz and 38.0 GHz are calculated for power input of 50 mW & 500 mW which is standard for SAR calculated by EM simulator. The value of input power is also mentioned in the revised manuscript and same has been highlighted. Also, Table 4 and Table 5 are added for more insight on SAR calculations

4. This paper designs a high-isolation, high-gain MIMO antenna, which effectively reduces inter-user interference. It is suggested that the authors strengthen the theoretical foundation supporting this conclusion by citing relevant works, such as “Achieving Positive Rate of Covert Communications Covered by Randomly Activated Overt Users” and “Energy Efficiency of Massive Random Access in MIMO Quasi-Static Rayleigh Fading Channels With Finite Blocklength”, to make the claims more convincing.

Response: Authors are thankful to reviewer for suggesting the quality papers addressing the covert communication and Rayleigh Fading Channels. The suggested research papers has been added in the revised manuscript and same has been highlighted.

5. Please pay more attention to writing conventions and consistency. For example, the reference citation format is inconsistent in the Introduction, where references [1] and [3] are used differently. In addition, some abbreviations (e.g., ECC) are not defined when they first appear in the text.

Response: Authors are thankful to reviewer for raising the concerns related to conventions. All the conventions are well defined in the revised manuscript and same has been highlighted.

---

## [Decision Letter · Decision Letter 1]

15 Jan 2026

Micromachined mmWave 28.0/38.0 GHz MIMO Antenna Loaded with Frequency Selective Surface for Gain Enhancement and SAR Analysis for Future Wireless Applications

PONE-D-25-52779R1

Dear Dr. Ali,

We’re pleased to inform you that your manuscript has been judged scientifically suitable for publication and will be formally accepted for publication once it meets all outstanding technical requirements.

Kind regards,

Neng Ye

Academic Editor

PLOS One

Additional Editor Comments (optional):

The authors have addressed all the comments. I'm OK with this version.

Reviewers' comments:

Reviewer's Responses to Questions

**Comments to the Author**

Reviewer #1: All comments have been addressed

Reviewer #2: All comments have been addressed

2. Is the manuscript technically sound, and do the data support the conclusions?

Reviewer #1: Yes

Reviewer #2: Yes

3. Has the statistical analysis been performed appropriately and rigorously?

Reviewer #1: Yes

Reviewer #2: Yes

4. Have the authors made all data underlying the findings in their manuscript fully available?

Reviewer #1: (No Response)

Reviewer #2: Yes

5. Is the manuscript presented in an intelligible fashion and written in standard English?

Reviewer #1: Yes

Reviewer #2: Yes

Reviewer #1: The authors have revised the comments carefully. The paper can be accepted in its current form…….…….

Reviewer #2: The authors have satisfactorily addressed all comments, and I am satisfied with this revised version.

**Do you want your identity to be public for this peer review?** For information about this choice, including consent withdrawal, please see our Privacy Policy

Reviewer #1: **Yes:** Ahmed Jamal Abdullah Al-Gburi

Reviewer #2: No

---

## [Editor Report · Acceptance letter]

PONE-D-25-52779R1

PLOS One

Dear Dr. Ali,

I'm pleased to inform you that your manuscript has been deemed suitable for publication in PLOS One. Congratulations! Your manuscript is now being handed over to our production team.

Kind regards,

on behalf of

Dr. Neng Ye

Academic Editor

PLOS One